# Long-term drought and risk of infant mortality in Africa: A cross-sectional study

Pin Wang[1,2,3]*, Tormod Rogne[4,5], Joshua L. Warren[5,6,7], Ernest O. Asare[7,8], Robert A. Akum[9], N'datchoh E. Toure[10], Joseph S. Ross[11,12,13], Kai Chen[1,2]

1 Department of Environmental Health Sciences, Yale School of Public Health, New Haven, Connecticut, United States of America, 2 Yale Center on Climate Change and Health, Yale School of Public Health, New Haven, Connecticut, United States of America, 3 Department of Global, Environmental, and Occupational Health, School of Public Health, University of Maryland, College Park, Maryland, United States of America, 4 Department of Chronic Disease Epidemiology, Yale School of Public Health, New Haven, Connecticut, United States of America, 5 Center for Perinatal, Pediatric and Environmental Epidemiology, Yale School of Public Health, New Haven, Connecticut, United States of America, 6 Department of Biostatistics, Yale School of Public Health, New Haven, Connecticut, United States of America, 7 Public Health Modeling Unit, Yale School of Public Health, New Haven, Connecticut, United States of America, 8 Department of Epidemiology of Microbial Diseases, Yale School of Public Health, New Haven, Connecticut, United States of America, 9 Department of Geography, SDD University of Business and Integrated Development Studies, Wa, Ghana, 10 LASMES, UFR SSMT University Felix Houphouet-Boigny Abidjan-Cocody, Abidjan, Côte d'Ivoire, 11 Section for General Medicine, Department of Internal Medicine, Yale School of Medicine, New Haven, Connecticut, United States of America, 12 Department of Health Policy and Management, Yale School of Public Health, New Haven, Connecticut, United States of America, 13 Center for Outcomes Research and Evaluation, Yale-New Haven Health System, New Haven, Connecticut, United States of America

* pinwang@umd.edu

## Abstract

### Background

As extreme events such as drought and flood are projected to increase in frequency and intensity under climate change, there is still large missing evidence on how drought exposure potentially impacts mortality among young children. This study aimed to investigate the association between drought and risk of infant mortality in Africa, a region highly vulnerable to climate change that bears the heaviest share of the global burden.

### Methods and findings

In this cross-sectional study, we obtained data on infant mortality in 34 African countries during 1992–2019 from the Demographic and Health Surveys program. We measured drought by the standardized precipitation evapotranspiration index at a timescale of 24 months and a spatial resolution of 10 × 10 km, which was further dichotomized into mild and severe drought. The association between drought exposure and infant mortality risk was estimated using Cox regression models allowing time-dependent covariates. We further examined whether the association varied for neonatal and post-neonatal mortality and whether there was a delayed association with drought exposure during pregnancy or infancy.

The mean (standard deviation) number of months in which children experienced any drought during pregnancy and survival period (from birth through death before 1 year of

**Data availability statement:** Data on infant mortality and socioeconomic status in this study were collected from the Demographic and Health Surveys program (https://dhspro-gram.com/), which are freely available upon request. Publicly available meteorological records were obtained from the fifth gener-ation ECMWF atmospheric reanalysis of the global climate (ERA5-Land, https://cds.climate.copernicus.eu/).

**Funding:** PW, EOA, and KC received support from the Reckitt Global Hygiene Institute (https://rghi.org/; RGHI RIN: 2021-001). TR was funded by CTSA (Grant Number UL1 TR001863) from the National Center for Advancing Translational Science (NCATS; https://ncats.nih.gov/), a component of the National Institutes of Health (NIH). JSR currently receives research support through Yale University from Johnson and Johnson to develop methods of clinical trial data sharing, from the Food and Drug Administration for the Yale-Mayo Clinic Center for Excellence in Regulatory Science and Innovation (CERSI; https://www.fda.gov/science-research/advancing-regulatory-science/centers-excel-lence-regulatory-science-and-innovation-cer-sis) program (U01FD005938), from the Agency for Healthcare Research and Quality (R01HS022882) (https://www.ahrq.gov/), and from Arnold Ventures; formerly received research support from the Medical Device Innovation Consortium as part of the National Evaluation System for Health Technology (NEST; https://www.fda.gov/about-fda/cdrh-re-ports/national-evaluation-system-health-tech-nology-nest) and from the National Heart, Lung and Blood Institute (https://www.nhlbi.nih.gov/) of the National Institutes of Health (NIH) (R01HS025164, R01HL144644). The funders had no role in study design, data collection and analysis, decision to publish, or preparation of the manuscript.

**Competing interests:** The authors have declared that no competing interests exist.

**Abbreviations:** CI, confidence interval; DHS, Demographic and Health Surveys; HR, hazard ratio; IMR, infant mortality rate; LMICs, low- and middle-income countries; SPEI, standardized precipitation evapotranspiration index; STROBE, Strengthening the Reporting of Observational Studies in Epidemiology.

age) was 4.6 (5.2) and 7.3 (7.4) among cases and non-cases, respectively. Compared to children who did not experience drought, we did not find evidence that any drought exposure was associated with an increased risk of infant mortality (hazard ratio [HR]: 1.02, 95% confidence interval [CI] [1.00, 1.04], $p = 0.072$). When stratified by drought severity, we found a statistically significant association with severe drought (HR: 1.04; 95% CI [1.01, 1.07], $p = 0.015$), but no significant association with mild drought (HR: 1.01; 95% CI [0.99, 1.03], $p = 0.353$), compared to non-exposure to any drought. However, when excluding drought exposure during pregnancy, the association with severe drought was found to be non-significant. In addition, an increased risk of neonatal mortality was associated with severe drought (HR: 1.05; 95% CI [1.01, 1.10], $p = 0.019$), but not with mild drought (HR: 0.99; 95% CI [0.96, 1.02], $p = 0.657$).

## Conclusions

Exposure to long-term severe drought was associated with increased infant mortality risk in Africa. Our findings urge more effective adaptation measures and alleviation strategies against the adverse impact of drought on child health.

## Author summary
### Why was this done?

- Climate change is projected to result in a higher frequency and intensity of drought events.
- Previous studies have found rainfall anomaly to be associated with increased infant mortality.
- The risk of drought exposure for infant health outcomes is not well understood.

### What did the researchers do and find?

- Employing an advanced measure of drought incorporating multiple climatic variables, we found no evidence that any drought exposure was associated with an increased risk of infant mortality in Africa.
- When stratifying the analysis by drought severity, exposure to long-term severe drought was observed to be associated with higher infant mortality risk.
- There is a delayed association between exposure to drought during pregnancy and risk of infant mortality.

### What do these findings mean?

- The protection of pregnant women and young children from drought could potentially help reduce the heavy toll of infant mortality in Africa.
- Our study urges the prioritization of neonatal and pediatric care as well as the development of water, sanitation, and hygiene infrastructure and adaptation strategies in rural Africa.
- This study may underestimate the potential impact of drought on mortality in early life given it could have already contributed to miscarriage or stillbirth.

## Introduction

Infant mortality rate (IMR), defined as the number of deaths among children younger than 1 year per 1,000 live births, is an important measure of overall population health, with global rates decreasing from 64.6 deaths per 1,000 live birth in 1990 to 28.4 deaths per 1,000 in 2021 [1]. Despite this trend towards improvement, the world is still struggling to achieve the Sustainable Development Goal 3.2 on reducing under-five child mortality [2], of which infant mortality is the largest contributor [3].

Environmental exposures such as polluted air [4], unsafe water [5], and unsatisfactory sanitation [6] have been associated with an increased risk of infant mortality. The risks associated with drought are less well understood. Drought is a complex environmental phenomenon influenced by multiple climatological parameters, with low precipitation as the main driver [7]. Worldwide, climate change is projected to increase the occurrences of climate extremes, particularly in Africa, identified as one of the regions highly vulnerable to these changes due to high exposure and low adaptive capacity [8], including projections of increased frequency, heightened intensity, and prolonged duration of drought hazards in many parts of the world [9]. Drought affects human health through many interacting pathways [10,11]. In particular, studies have observed drought to be associated with water-borne [12] and vector-borne [13] diseases, cardiovascular [14] and respiratory [15] diseases, malnutrition [16], mental disorders [17], and mortality [18–23]. To our knowledge, only one previous study in Ethiopia looked into the quantitative impact of drought on under-five child mortality, with no significant result reported [24]. Furthermore, there is a paucity of evidence on the association between drought and infant mortality, although two studies from China and Brazil suggested such a link, in which drought exposure was characterized by rainfall fluctuations during the 12 months before birth [25,26]. Despite the lack of consensus on drought definition [27], there has been a growing recognition of the significance of incorporating temperature and other meteorological information, simultaneously taking water supply and demand into account when computing drought indices [7]. However, previous studies only included precipitation in drought characterization [25,26] and evidence on how such a more rigorously defined drought measure is associated with risk of infant deaths is missing.

In addition, over half of infant mortality is attributed to deaths during the first 27 days after birth [3]. However, it remains unclear whether exposure to drought impacts risk of neonatal and post-neonatal mortality differently. Furthermore, the indirect affecting pathway of drought on the infant through the mother during pregnancy may delay its association with subsequent deaths during infancy. However, the evidence on the potentially lagged association between drought and infant mortality is lacking.

To date, there is a scarcity of research investigating the relationship between drought and infant mortality in Africa, a region that bears the greatest burden of infant deaths worldwide [1] and where some profound drought events have occurred in the last century [28]. In Africa, infant mortality is associated not only with various demographic and socioeconomic characteristics, but also with local cultural settings and societal structural frameworks, such as differences in healthcare facilities provision between urban and rural areas, domestic healthcare expenditure, and foreign resources flowing into health [29,30], which remain challenges the continent still faces. In addition, this region of the world deserves careful scrutiny, not only because the human health effects of climate change are infrequently studied among African populations [31], but also because it is one of the fastest growing populations, expected to be home to a quarter of the world's population by 2050 [32].

To fill these research gaps, by using data from 34 African countries and developing a complex drought indicator, we aimed to (1) examine the association between drought and risk of infant mortality; (2) investigate whether the association varies between neonatal and

post-neonatal deaths; and (3) explore the potential delayed effect of drought exposure on infant mortality.

## Methods

### Study population and infant mortality

The Demographic and Health Surveys (DHS) program routinely conducts nationally representative household surveys in over 90 low- and middle-income countries (LMICs) [33]. We obtained individual-level mortality data of children born during 1992–2019 within the 5 years prior to survey administration in 34 African LMICs. Using a two-stage sampling process, DHS selects survey clusters stratified by geographic region and urban/rural residence, and interviews 20–30 randomly selected households per cluster [33]. Mothers of reproductive age (15–49 years) in these sampled households were queried about each birth during the past 5 years. The geocoded coordinate location of each survey cluster was recorded as the centroid of the interviewed households, which was randomly displaced by 2–10 km to protect anonymity by DHS design [34]. To compute a wealth index that is comparable across surveys, we also extracted data on household characteristics, including source of drinking water, type of toilet facilities, electricity, type of flooring, and possession of radio, television, phone (landline or cellphone), refrigerator, motorcycle, and car. Following the method described by Bendavid and colleagues [35], we then performed a principal component analysis, generated a wealth index, and employed the quintiles of this index in the subsequent analysis.

### Drought and other climate data

We quantified long-term drought conditions by the standardized precipitation evapotranspiration index (SPEI), which offers a robust and reliable measure to account for both water supply and demand [7], at a timescale of 24 months. We have previously described in detail how to calculate drought indicators [12]. Briefly, we first downloaded gridded monthly meteorological data at a resolution of 0.1° (~10 × 10 km) during the study period from the fifth-generation European Centre for Medium-Range Weather Forecasts atmospheric reanalysis of the global climate (ERA5-Land). We then calculated the climatic water balance as the difference between the available water content of soil and vegetation, measured by precipitation, and potential evapotranspiration, calculated using maximum and minimum temperatures, dew point temperature, wind speed, solar radiation, air pressure, latitude, and elevation. The SPEI is a monthly measure accounting for the overall cumulative effects of climatic conditions during a preceding period (S1 Fig). In order to compute the SPEI at a timescale of 24 months (referred to as SPEI-24 hereinafter) for a given month during the study period, we first generated a time series by summing the water balance over the preceding 23 months up to the current month; then we transformed this series to a normal distribution with a mean of zero and standard deviation of one, according to a log-logistic distribution, to eliminate the effect of both the local season and climate patterns [36].

Next, we linked each child with the gridded monthly SPEI, mean temperature, and total rainfall based on the geographic location of each cluster and the month and year of the child's birth. Therefore, the exposure period was defined as from the start of pregnancy to death or the 12th month after birth, depending on whether a child died before 1 year of age (every pregnancy was assumed to be full-term [9 months] due to largely missing information on gestational age). Finally, adapting from the classification from the Federal Office of Meteorology and Climatology MeteoSwiss, we defined three types of drought events based on their severity (any drought: SPEI ≤ −0.5; mild drought: −1.3 < SPEI ≤ −0.5; severe drought: SPEI ≤ −1.3) [12].

## Statistical analysis

We used an extended Cox regression model with time-dependent covariates to examine the association between drought at a timescale of 24 months and risk of infant mortality. Previously applied in other infant death studies [37], this extended model allows for time-varying covariance caused by the change of a given covariate over time during the follow-up period [38]. It is necessary to reiterate that our explanatory variable, the monthly drought indicator, changed over time (indicating the presence or absence of drought conditions for each month from conception to infant death). Our time-dependent Cox regression analysis was an appropriate approach for handling the time-varying drought indicator because, during each month in which an infant death occurred, the model compared the current drought condition experienced by the deceased infant with the current drought conditions experienced by all other infants at risk at that time [39]. With its ability to incorporate time-varying covariates, this method has been widely adopted in both environmental and non-environmental epidemiological research [37,40].

Specifically, we included in the model a time-dependent indicator for drought exposure and a series of time-invariant covariates, including child's sex, area of residence, mother's education, and wealth quintile. Categorical birth month, a natural cubic spline of birth year with three degrees of freedom, and a random intercept for a composite indicator for country and survey cluster were also included to adjust for seasonality, long-term trend, and cross-location differences, respectively (S1 Method). To explore whether risk of neonatal and post-neonatal mortality was differently associated with drought exposure, we applied the same model for infant deaths during the first 27 days and from the 28th day through the 12th month after birth.

We then stratified the main analysis by several baseline characteristics by incorporating an interaction term between the binary indicator for any drought and each of the following factors: child's sex (male and female), area of residence (urban and rural), mother's education (no education and any education), household's wealth status (lower as the 1st–2nd quintiles and higher as the 3rd–5th quintiles), year period of birth (1992–2005 and 2006–2019), and climate zone (tropical, temperate, and dry). The statistical significance of the difference between subgroups was indicated by the $p$-value of the interaction term.

We then performed several secondary analyses. We first evaluated the association of drought exposure with infant mortality by month of death (27 days after birth for neonatal mortality), for which two types of exposure were assessed, namely a monthly binary indicator and the number of drought months experienced during pregnancy or infancy. Second, we combined all post-neonatal deaths (between 28 and 364 days of age) and examined whether neonatal and post-neonatal mortality were differently associated with monthly exposure during pregnancy. With the same exposure assessment period for all infant deaths within a specific month, we employed mixed-effects logistic regression models to quantify these associations (S1 Method). We categorized children who died during a specific month/period as cases and children who survived beyond that month/period as non-cases. For the death-month-specific analysis, we excluded children who were alive but younger than the age of that month at the time of the interview. Additionally, children who had died before that month were accounted for in the models that examined the association in previous months, and thus were also excluded from the analysis for the relevant month.

Several sensitivity analyses were conducted to test the model robustness. First, we separately included a natural cubic spline of monthly total precipitation with three degrees of freedom as a time-varying covariate in the main model. Second, we removed the month of birth from the main model to evaluate the impact of seasonality. Third, we adopted a discrete-time complementary log–log binomial regression model given the similar relative hazard interpretation of its coefficients to the Cox regression model in time-to-event analysis [41].

Fourth, we performed an additional analysis to compare the association with exposure from birth through infant deaths to the association with exposure from the start of pregnancy through infant deaths (main analysis). In addition, to test the importance of including gestational exposure in our main model, we incorporated drought during pregnancy in the fourth sensitivity analysis starting from birth. Since there were nine drought indicators during pregnancy for each child, we computed two alternative variables to assess gestational drought in the models: (1) an indicator of the total number of drought months a child experienced during pregnancy as a continuous variable, and (2) an indicator of any drought month a child ever experienced during pregnancy as a binary variable.

Only any drought was evaluated in the subgroup analysis and secondary analysis. We applied DHS sampling weight in all models to take into account the influence of survey representations [42]. The significance level for all statistical tests was set at two-sided $p <$ 0.05. While this study did not have a prospectively registered protocol and statistical analysis plan, all variable definitions and analyses were pre-specified over the course of communications between the study team members and during laboratory meetings. All data analyses were completed using R statistical software 4.3.1, with the *SPEI* package for drought exposure assessment, and the *coxme* and *glmmTMB* packages for regression analysis.

This study is reported as per the Strengthening the Reporting of Observational Studies in Epidemiology (STROBE) guideline (S1 STROBE Checklist). The Yale Institutional Review Board determined this study as not-human-subject research (IRB ID: 2000036064); thus, ethics approval for this study was not required.

## Results

Our study included a total of 850,924 children from 103 surveys conducted in 34 countries during 1992–2019. The overall IMR in surveyed participants in Africa was 59.2 per 1,000 live births (50,377 deaths) (Table 1), with the highest observed in Sierra Leone (82.3), followed by Mali (82.0) and Eswatini (80.8), and the lowest in Gabon (27.5) (Fig 1A). The average (standard deviation) count of months in which children encountered all, mild, or severe drought during both pregnancy and the survival period was as follows among cases: 4.6 (5.2), 3.3 (4.1), or 1.3 (3.1), respectively. In comparison, among non-cases (i.e., children survived 12 months after birth and alive infants at the time of interview), the respective number of months was 7.3 (7.4), 5.2 (5.6), or 2.0 (4.3). The spatial distribution of drought events exhibits significant variation across the continent (Fig 1B). Eswatini experienced the highest average number of drought months (135.5 months) (area-weighted mean across all 10-km grids), whereas these events were least frequent in Guinea (39.9 months) (Fig 1C). In addition, fewer mothers and children experienced severe droughts than mild droughts, and more drought events were observed for children born during 2006–2019 than for children born during 1992–2005 and for those living in the tropical zone than for those living in the temperate or dry zone (S1 Table). The number of infant deaths during the first 27 days was significantly higher than the following 11 months (S2 Table).

Compared to children who did not experience drought, we did not find evidence that any drought exposure was associated with an increased risk of infant mortality (hazard ratio [HR]: 1.02; 95% confidence interval [CI] [1.00, 1.04], $p = 0.072$) (Fig 2). When the analysis was stratified by drought severity, a significant positive association was observed for severe drought (HR: 1.04; 95% CI [1.01, 1.07], $p = 0.015$), whereas a null association was observed for mild drought (HR: 1.01; 95% CI [0.99, 1.03], $p = 0.353$). This null association with mild drought was estimated for both post-neonatal mortality (HR: 1.02; 95% CI [0.99, 1.06], $p = 0.203$) and neonatal mortality (HR: 0.99; 95% CI [0.96, 1.02], $p = 0.657$). However, severe drought was

**Table 1. Study population characteristics in 34 low- and middle-income African countries during 1992–2019 by baseline characteristics (total N = 850,924).**

| | Missing (%) | Infant mortality (N [%]) | | Infant mortality rate (per 1,000 live births) | |
|---|---|---|---|---|---|
| | | Case | Non-case | Unweighted | Weighted |
| Total | | 50,377 | 800,547 | 59.2 | 59.2 |
| Child's sex | 0 | | | | |
| Male | | 27,802 (55.2) | 403,017 (50.3) | 64.5 | 64.7 |
| Female | | 22,575 (44.8) | 397,530 (49.7) | 53.7 | 53.7 |
| Mother's education | 0 | | | | |
| No education | | 25,973 (51.6) | 370,310 (46.3) | 65.5 | 65.2 |
| Primary | | 16,422 (32.6) | 266,580 (33.3) | 58.0 | 58.3 |
| Secondary | | 7,310 (14.5) | 145,469 (18.2) | 47.8 | 47.7 |
| Higher | | 671 (1.3) | 18,133 (2.3) | 35.7 | 41.0 |
| Residence | 0 | | | | |
| Urban | | 11,563 (23.0) | 216,519 (27.0) | 50.7 | 48.4 |
| Rural | | 38,814 (77.0) | 584,028 (73.0) | 62.3 | 63.2 |
| Wealth quintile | 11.1 | | | | |
| Lowest | | 11,093 (25.3) | 153,525 (21.5) | 67.4 | 67.3 |
| Second | | 9,529 (21.8) | 135,105 (19.0) | 65.9 | 66.7 |
| Middle | | 8,534 (19.5) | 140,712 (19.7) | 57.2 | 57.2 |
| Fourth | | 8,287 (18.9) | 143,021 (20.1) | 54.8 | 52.9 |
| Highest | | 6,368 (14.5) | 140,505 (19.7) | 43.4 | 44.5 |
| Year of birth | 0 | | | | |
| 1992–2005 | | 20,615 (40.9) | 253,104 (31.6) | 75.3 | 76.6 |
| 2006–2019 | | 29,762 (59.1) | 547,443 (68.4) | 51.6 | 51.0 |
| Climate zone | 0 | | | | |
| Tropical | | 30,515 (60.6) | 472,541 (59.0) | 60.7 | 58.7 |
| Temperate | | 7,633 (15.2) | 130,152 (16.3) | 55.4 | 65.2 |
| Dry | | 12,229 (24.3) | 197,854 (24.7) | 58.2 | 56.5 |

found to be significantly associated with neonatal mortality (HR: 1.05; 95% CI [1.01, 1.10], $p = 0.019$), but not with post-neonatal mortality (HR: 1.04; 95% CI [0.99, 1.09], $p = 0.120$), compared to non-exposure to any drought.

We found that the association was significantly stronger among children living in rural areas than those living in urban areas and among children living in tropical climate zones than those living in dry zones (Fig 3). We did not observe significant effect modification by other factors.

We found the association of monthly exposure to drought during pregnancy with neonatal mortality to be non-significant except for that during the 1st month of pregnancy (odds ratio: 1.04; 95% CI [1.01, 1.07], $p = 0.012$) (Figs 4A and S1, 9th month before birth). Furthermore, monthly drought exposure was associated with infant mortality risk during the 7th–8th month after birth in a more consistent and significant fashion (Fig 4A), and the significant increase in mortality risk was observed in the 3rd and 7th–8th month for the number of drought months during pregnancy and only in the 7th month for the number of drought months during infancy (Fig 4B). We also found that the association with all post-neonatal mortality was observed to be significant during the 2nd–4th and 8th–9th month of pregnancy (S2 Fig, 6th–8th and 1st–2nd month before birth).

The results from the sensitivity analysis suggested that our estimation was largely robust, although we observed slight attenuation of the association with all types of drought when

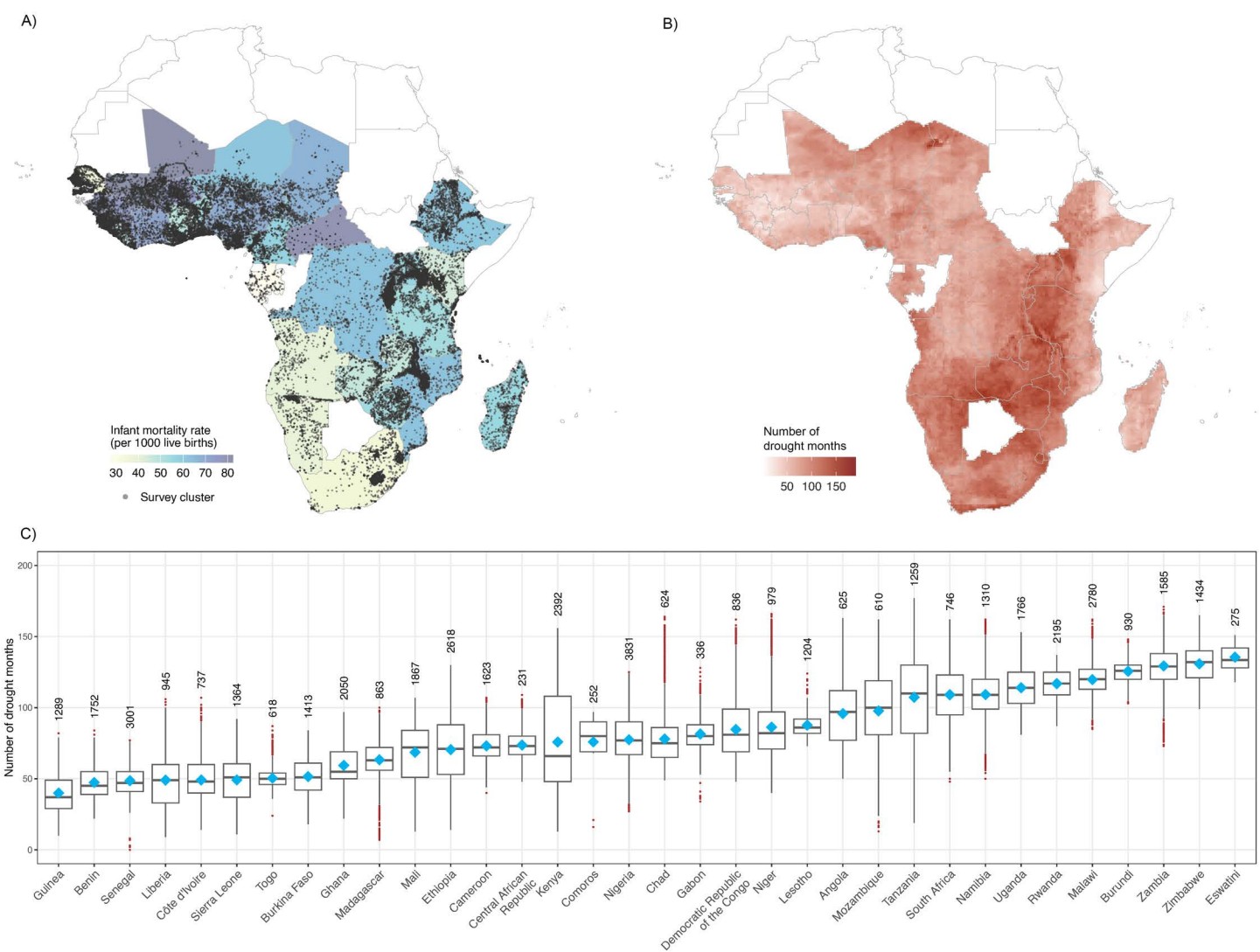

**Fig 1. Infant mortality rate (deaths per 1,000 live births) (A), number of drought months (B), and country-specific distributions of the number of drought months across all 10-km grids, sorted by area-weighted mean value (diamonds) and labeled with number of survey clusters (C) in Africa during 1992–2019.** The base layer of the map was obtained from ArcGIS Hub (https://hub.arcgis.com/datasets/07610d73964e4d39ab62c4245d548625/).

applying complimentary log–log binary regression models (S3 Table). In addition, when we started the analysis from birth, the association with severe drought was found to be non-significant (S3 Table). However, when we included drought exposure during pregnancy as a covariate in this postnatal-period-only analysis, it showed a statistically significant association with our outcome (S4 Table), further demonstrating the importance of including this prenatal exposure to drought in our primary analysis.

## Discussion

In this cross-sectional, multi-country study, we used an advanced drought measure and found no evidence that any drought exposure was associated with an increased risk of infant mortality in African LMICs. However, long-term severe drought exposure was observed to be significantly associated with an increased risk of infant mortality. A significantly stronger

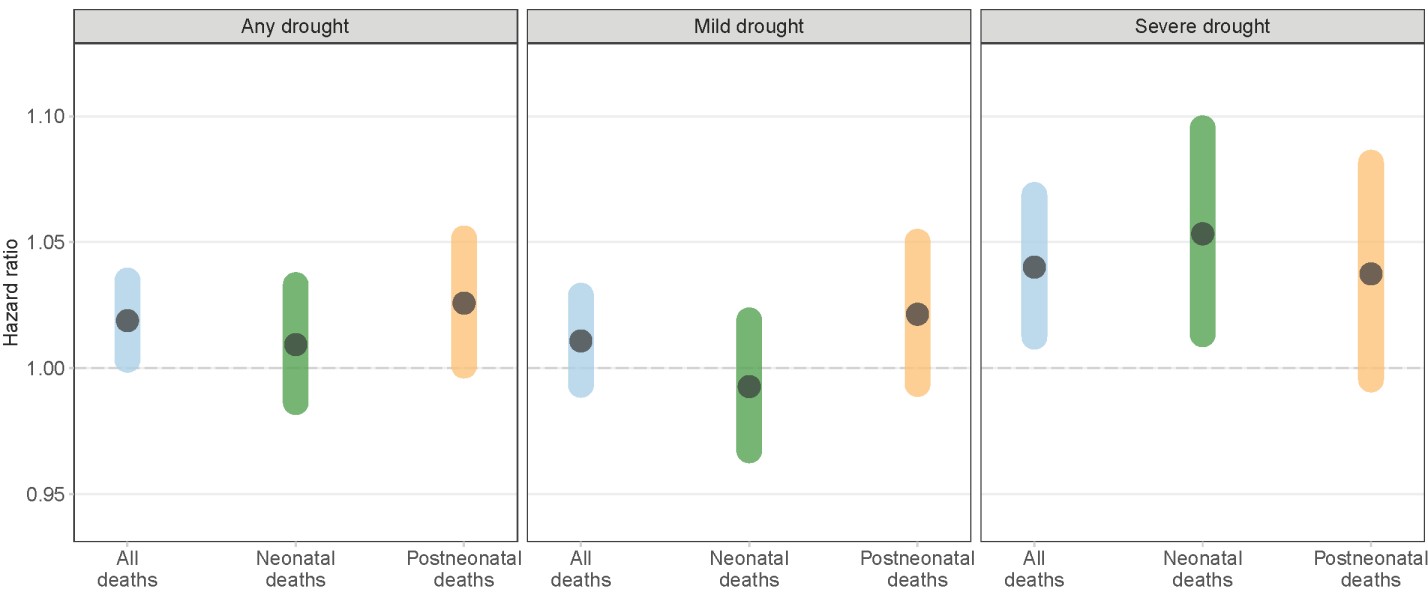

**Fig 2. Error bar charts for the associations between risk of infant mortality and exposure to long-term drought represented by 24-month standardized precipitation and evapotranspiration index by drought severity (any, mild, or severe) and type of infant mortality (neonatal or post-neonatal mortality).** Reference group is the children who did not experience droughts. Covariates in the models included child's sex, area of residence, mother's education, and wealth quintile, categorical birth month, a natural cubic spline of birth year with three degrees of freedom, and a random intercept for a composite indicator for country and survey cluster.

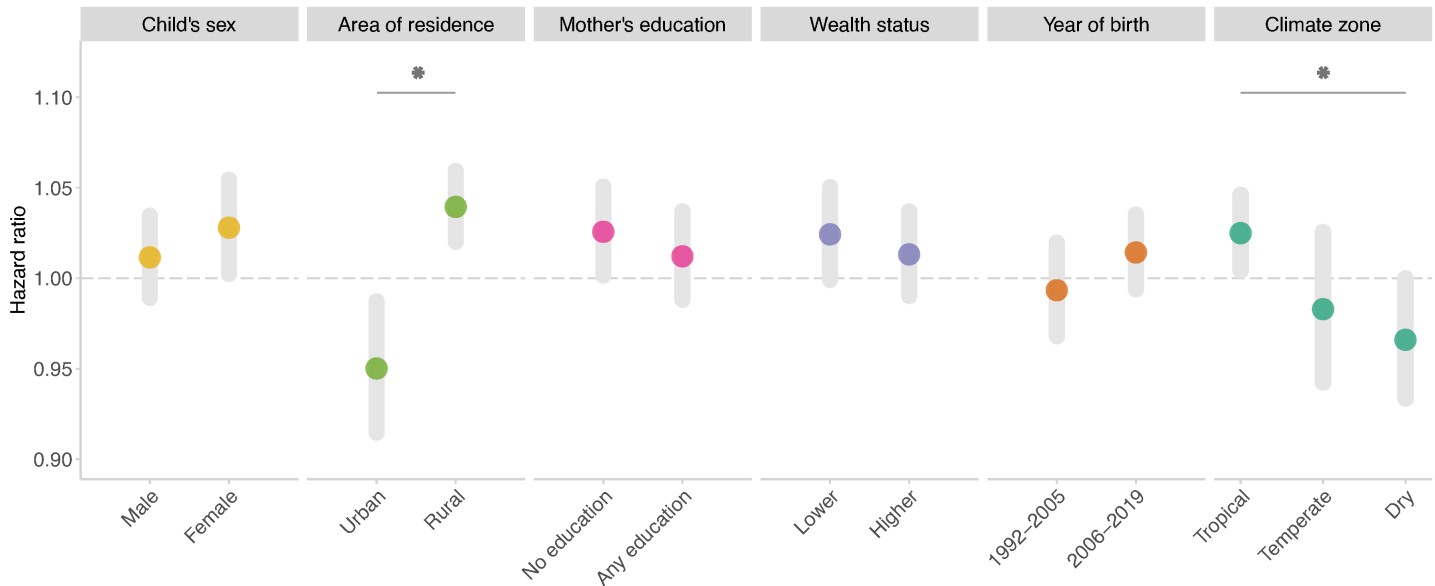

**Fig 3. Associations between risk of infant mortality and exposure to long-term drought stratified by baseline characteristics, year period of birth, and climate zone.** Statistically significant pairwise differences ($p < 0.05$) are marked with an asterisk.

association was observed among children living in rural areas than among those in urban areas, and among children living in tropical zones compared to those in dry zones.

Researchers have investigated the relationship between drought and under-five child mortality, with no plausible association observed [24]. Compared to child mortality during

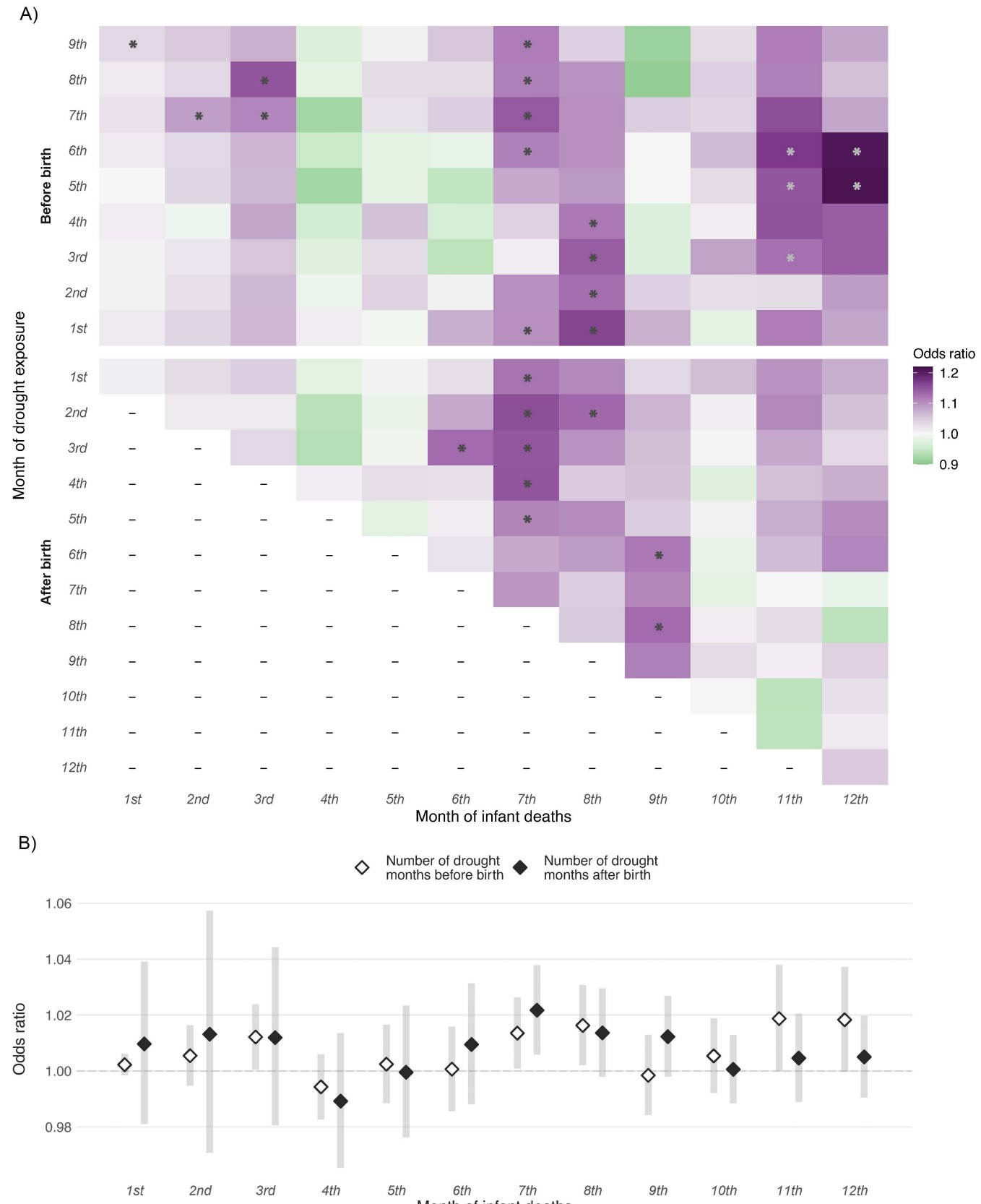

**Fig 4. Associations between infant deaths during each month within 1 year of age and long-term drought.** The exposure was measured by a binary drought indicator during each month before and after birth (the ninth and first month of drought exposure before birth represents the first and last month

of pregnancy, respectively) **(A)** and by the number of drought months experienced before and after birth (from the first through the last month of pregnancy for prenatal exposure and from the month of birth through the relevant month examined for postnatal exposure) **(B)**. Statistically significant results ($p <$ 0.05) are marked with an asterisk.

1–4 years after birth, infant mortality, the largest contributor of under-five mortality [3], is more likely to be biologically linked to drought exposure during pregnancy. However, there is still limited evidence on the negative impact of drought on infant deaths. Two studies in China and Brazil defined drought as rainfall shocks or fluctuations, which were calculated as the log deviation of total rainfall during the 12 months before birth relative to the historical average yearly total rainfall, and both studies reported increased risk of infant mortality associated with rainfall deficit [25,26]. However, no other climatic conditions were involved in the drought indicator calculation. Similarly, a global study employed positive or negative rainfall shocks, defined as one standard deviation above or below the historical average rainfall, and found a negative or positive association with IMR, respectively [43]. Our study, for the first time, assessed the relationship between infant mortality and drought considering both water supply and demand, and also found a heightened risk associated with long-term drought conditions. Moreover, by using a monthly drought indicator, instead of annual rainfall deficit employed by previous studies, our study was able to examine the critical window of the association between drought exposure and infant mortality.

The exact mechanism behind this association is still unknown. However, the interaction among drought, water, sanitation, and hygiene practices, and diarrhea may play a notable role in the effect pathway. Drought has been found to be associated with an increased risk of diarrhea [12], one of the leading causes of death in children under five. In addition, drought not only directly declines surface water and groundwater quantity through reduced precipitation and enhanced evapotranspiration, but also deteriorates water quality of freshwater systems via increased salinity, turbidity, algal level, and metal and pathogen concentration [44]. Inadequate access to water and consumption of unsafe drinking water could either directly link to infant deaths [45,46], or increase the likelihood of diarrhea, subsequently contributing to higher mortality. Furthermore, drought has a detrimental effect on crop yield and livestock, and plays a substantial role in driving land degradation and potentially triggering forced migration [47], posing a threat of food insecurity, which in turn leads to undernutrition for pregnant women and infant—an additional contributor to infant mortality. Specifically, suboptimal nutrition during pregnancy increases the risk of low birth weight, a well-documented factor in infant deaths [48]. In addition, poor nutritional status among children, characterized by underweight, stunting, and wasting, can also contribute to the burden of infant mortality [49].

There could be delays from drought exposure to diarrhea occurrences or poor water, sanitation, and hygiene practices, both of which could also directly or indirectly result in infant deaths. Indeed, we found the association of in-utero or postnatal drought exposure to be lagged for several months in our month-specific analysis. However, the significant association we observed between severe drought and neonatal mortality implied the stronger and more acute impact of long-term drought condition with greater intensity. To our knowledge, this is the first study looking into the delayed association between rigorously defined drought measure and infant health. Apart from the direct impact due to insufficient water supply, all indirect pathways linking drought to infant mortality contribute to the time lag in the association. Additionally, younger infants are likely to be less vulnerable due to breastfeeding, which may explain the null associations we observed for the first 2 months after birth. However, the reason behind the exhibited lag pattern (elevation in the association during the 3rd, 7th–8th,

and 11th–12th month) is unknown and future studies are needed to further investigate the delayed effect of drought.

We observed a significantly stronger association among children residing in tropical zones than those living in dry zones. Interestingly, the association between drought exposure and diarrhea among children in tropical zones was found to be significantly weaker or null in our previous work [12]. The reason behind this discrepancy remains unclear. However, this contrast could stem from the more indirect and lagged influence of drought on infant mortality. With a delayed association, despite greater resources, tropical zones may still lack the capacity to adapt to more persistent drought conditions, compared to temperate and dry zones that have already learned to adapt to intermittent drought conditions. Rural areas were found to bear a higher burden of the impact from drought events than urban areas. Maternal and child healthcare facilities are likely to be absent or significantly fewer in rural settings, and poor road networks in these areas would further hinder accessibility to these facilities. Furthermore, there may be insufficient water, sanitation, and hygiene practices in the traditional childbirth environment in rural African regions [50,51], and drought exposure could further diminish the already inadequate washing practices, posing notably increased risk of life-threatening infections among newborns.

We acknowledge several limitations. First, our survival analysis started at the beginning of pregnancy, whereas our outcome of interest was infant mortality, which could not occur until birth. This misalignment may introduce immortal time bias, potentially over- or underestimating our investigated association. Second, despite a monthly measure, the calculated 24-month SPEI in our study incorporates multiple weather parameters to illustrate the overall dry condition for the preceding 24 months. However, drought and other unfavorable climatic conditions during this period could have already resulted in deaths in an earlier stage (i.e., miscarriage or stillbirth), introducing live-birth bias [52]. This study focusing on infant mortality might potentially underestimate the detrimental impact of drought on deaths in early life. Third, DHS randomly displaced all geocoded survey cluster locations by 2–10 km to protect privacy [34], and thus our drought exposure may be misclassified, particularly for children living in a cluster close to a border of a 10-km grid. Fourth, the definition of our drought indicator and its severity classification was derived from a single source, which may vary across locations and, as a result, may potentially limit the generalizability of our findings. Fifth, due to largely missing information on gestational age, we assumed the same pregnancy duration (i.e., 9 months) for all live births, which may also lead to drought exposure misclassification. Sixth, in our secondary analysis we examined the associations between infant deaths during each of the 12 months after birth and drought exposure during each of the 21 months before and after birth, which might increase the possibility of chance associations due to multiple comparisons. Last, nutritional status and diarrhea occurrences both significantly impact infant mortality risk. However, we were not able to unravel the interrelationship between these factors and drought since DHS only queries these variables for living children.

In summary, this study found that exposure to long-term severe drought was associated with increased risk of infant mortality. Comprehensive engagement, planning, and preparedness should be advanced among pregnant women and children that are disproportionately affected by drought, particularly in rural and tropical settings. Understanding the association between drought and infant mortality can guide a multi-faceted approach in Africa involving early warning systems, neonatal and pediatric care infrastructure planning, community education, food security program development, and water and sanitation interventions. With more frequent and intense drought events projected due to climate change, coupled with inadequate drought adaptation capacity in LMICs, the protection of pregnant women and young children from drought could potentially play a vital role in alleviating the substantial burden of infant mortality in Africa.

## Supporting information

**S1 STROBE Checklist. STROBE Statement—Checklist of items that should be included in reports of cross-sectional studies.**
(DOCX)

**S1 Method. Model specification of main and secondary analysis.**
(DOCX)

**S1 Table. Descriptive statistics for the number of drought months experienced by all included children by exposure window, year of birth, climate zone, and drought severity.**
(DOCX)

**S2 Table. Number of infant deaths (among 850,924 children) during each month within 1 year of age and number of months for SPEI-24 any drought experienced before and after birth.** SPEI: standardized precipitation evapotranspiration index.
(DOCX)

**S3 Table. Association between long-term drought and risk of infant mortality in various sensitivity analyses.**
(DOCX)

**S4 Table. Estimates and uncertainties of drought exposure before and after birth when including gestational drought exposure in the sensitivity analysis of postnatal period only.**
(DOCX)

**S1 Fig. Illustration of calculation of SPEI at a timescale of 24 months for a given month from the start of pregnancy through death or the 12th month after birth.** Climate variables include maximum and minimum temperatures, dew point temperature, wind speed, solar radiation, and air pressure. SPEI: standardized precipitation evapotranspiration index.
(DOCX)

**S2 Fig. Associations between neonatal and post-neonatal mortality and drought exposure by month of pregnancy.**
(DOCX)

## Acknowledgments

The authors would like to thank the DHS Program, ICF International for providing the data used in the analysis. The views expressed in this paper are those of the authors and not necessarily those of RGHI. The contents are solely the responsibility of the authors and do not necessarily represent the official views of NIH. JSR was an expert witness at the request of Relator's attorneys, the Greene Law Firm, in a qui tam suit alleging violations of the False Claims Act and Anti-Kickback Statute against Biogen that was settled September 2022.

## Author contributions

**Conceptualization:** Pin Wang, Kai Chen.

**Data curation:** Pin Wang, Kai Chen.

**Formal analysis:** Pin Wang.

**Funding acquisition:** Tormod Rogne, Joseph S. Ross, Kai Chen.

**Investigation:** Pin Wang.

**Methodology:** Pin Wang, Tormod Rogne, Joshua L. Warren, Ernest O. Asare, Robert A. Akum, N'datchoh E. Toure, Joseph S. Ross, Kai Chen.

**Project administration:** Kai Chen.

**Supervision:** Kai Chen.

**Validation:** Pin Wang.

**Visualization:** Pin Wang.

**Writing – original draft:** Pin Wang.

**Writing – review & editing:** Pin Wang, Tormod Rogne, Joshua L. Warren, Ernest O. Asare, Robert A. Akum, N'datchoh E. Toure, Joseph S. Ross, Kai Chen.

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
