## [Editor Report · Decision Letter 0]

14 Mar 2024

Dear Dr Wang,

Thank you for resubmitting your manuscript entitled "The drier, the deadlier: The association between long-term drought and risk of infant mortality in Africa" for consideration by PLOS Medicine.

Your manuscript and rebuttal letter has now been evaluated by the PLOS Medicine editorial staff and I am pleased to let you know that we would like to send your revised submission out for external peer review.

Please re-submit your manuscript within two working days, i.e. by Mar 18 2024 11:59PM.

Feel free to email me at kjanin@plos.org if you have any queries relating to your submission.

Kind regards,

Katrien G. Janin, PhD

Senior Editor

PLOS Medicine

---

## [Decision Letter · Decision Letter 1]

18 Apr 2024

Dear Dr. Wang,

Thank you very much for submitting your manuscript "The drier, the deadlier: the association between long-term drought and risk of infant mortality in Africa" (PMEDICINE-D-24-00825R1) for consideration at PLOS Medicine.

As you will see, the reviewers were very positive about the paper and the importance of these follow-up data, but they raised a number of questions about specific study details and presentation. After discussing the paper with the editorial team and an academic editor with relevant expertise, I’m pleased to invite you to revise the paper in response to the reviewers’ comments. We plan to send the revised paper to some of all of the original reviewers*, and of course we cannot provide any guarantees at this stage regarding publication.

When you upload your revision, please include a point-by-point response that addresses all of the reviewer and editorial points, indicating the changes made in the manuscript and either an excerpt of the revised text or the location (eg: page and line number) where each change can be found. Please submit a clean version of the paper as the main article file and a version with changes marked should as a marked-up manuscript. Please also check the guidelines for revised papers at http://journals.plos.org/plosmedicine/s/revising-your-manuscript for any that apply to your paper.

We ask that you submit your revision by 9th of May. However, if this deadline is not feasible, please contact me by email, and we can discuss a suitable alternative.

Don’t hesitate to contact me directly with any questions (kjanin@plos.org). If you reply directly to this message, please be sure to ‘Reply All’ so your message comes directly to my inbox.

Kind regards,

Katrien

Katrien G. Janin, PhD

Associate Editor

PLOS Medicine

plosmedicine.org

Data Availability: 

PLOS Medicine requires that the de-identified data underlying the specific results in a published article be made available, without restrictions on access, in a public repository or as Supporting Information at the time of article publication, provided it is legal and ethical to do so. Please see the policy at http://journals.plos.org/plosmedicine/s/data-availability and FAQs at http://journals.plos.org/plosmedicine/s/data-availability#loc-faqs-for-data-policy   

PLOS defines the “minimal data set” to consist of the data set used to reach the conclusions drawn in the manuscript with related metadata and methods, and any additional data required to replicate the reported study findings in their entirety. Authors do not need to submit their entire data set, or the raw data collected during an investigation. 

For each data source used in your study:  

Financial Disclosure

The funding statement should include: specific grant numbers, initials of authors who received each award, URLs to sponsors’ websites. Also, please state whether any sponsors or funders (other than the named authors) played any role in study design, data collection and analysis, the decision to publish, or preparation of the manuscript. If they had no role in the research, include this sentence: “The funders had no role in study design, data collection and analysis, decision to publish, or preparation of the manuscript.

Reporting guidance 

Please report your study according to the relevant guidance which can be found here https://www.equator-network.org/reporting-guidelines/ 

Given your study is a observational cross-sectional study, Please ensure that the study is reported according to the STROBE guideline, and include the completed STROBE checklist as Supporting Information. Please add the following statement, or similar, to the Methods: ""This study is reported as per the Strengthening the Reporting of Observational Studies in Epidemiology (STROBE) guideline (S1 Checklist).The STROBE guideline can be found here: http://www.equator-network.org/reporting-guidelines/strobe/

Did your study have a prospective protocol or analysis plan? Please state this (either way) early in the Methods section.

For all observational studies, in the manuscript text, please indicate: (1) the specific hypotheses you intended to test, (2) the analytical methods by which you planned to test them, (3) the analyses you actually performed, and (4) when reported analyses differ from those that were planned, transparent explanations for differences that affect the reliability of the study's results. If a reported analysis was performed based on an interesting but unanticipated pattern in the data, please be clear that the analysis was data-driven.

Statistical reporting 

As per standard PLOS Medical requirements, please provide 95% CIs and p values for all results where appropriate (including the abstract). We suggest reporting statistical information in the following format: ‘x’; (95% CI [‘y’,’ z’] p value) and use commas as opposed to hyphens (as these can be confused with negative values) to separate upper and lower bounds. For p values, please report as p<0.001 and where higher as 'p=0.002'. Please add the statistical method used to your method section. We also invite you to report p values to consistently to the third decimal digit - thousandths.

If applicable, please include any important dependent variables that are adjusted for in the analyses.

Please include the actual amounts and/or absolute risk(s) of relevant outcomes (including NNT or NNH where appropriate), not just relative risks or correlation coefficients. (example for absolute risks: PMID: 28399126). 

Title 

Please revise your title according to PLOS Medicine's style. Your title must be nondeclarative and not a question. It should begin with main concept if possible. "Effect of" should be used only if causality can be inferred, i.e., for an RCT. Please place the study design ("A randomized controlled trial," "A retrospective study," "A modelling study," etc.) in the subtitle (ie, after a colon). 

Abstract layout 

Please structure your abstract using the PLOS Medicine headings (Background, Methods and Findings, Conclusions). 

Author summary 

At this stage, we ask that you include a short, non-technical Author Summary of your research to make findings accessible to a wide audience that includes both scientists and non-scientists. The authors summary should consist of 2-3 succinct bullet points under each of the following headings: 

Why Was This Study Done? Authors should reflect on what was known about the topic before the research was published and why the research was needed. 

What Did the Researchers Do and Find? Authors should briefly describe the study design that was used and the study’s major findings. Do include the headline numbers from the study, such as the sample size and key findings.     

What Do These Findings Mean? Authors should reflect on the new knowledge generated by the research and the implications for practice, research, policy, or public health. Authors should also consider how the interpretation of the study’s findings may be affected by the study limitations. In the final bullet point of ‘What Do These Findings Mean?’, please describe the main limitations of the study in non-technical language. 

The Author Summary should immediately follow the Abstract in your revised manuscript. This text is subject to editorial change and should be distinct from the scientific abstract. Please see our author guidelines for more information: https://journals.plos.org/plosmedicine/s/revising-your-manuscript#loc-author-summary 

Introduction layout 

Please address past research and explain the need for and potential importance of your study. Indicate whether your study is novel and how you determined that. If there has been a systematic review of the evidence related to your study (or you have conducted one), please refer to and reference that review and indicate whether it supports the need for your study. 

Discussion layout 

Please present and organize the Discussion as follows: a short, clear summary of the article's findings; what the study adds to existing research and where and why the results may differ from previous research; strengths and limitations of the study; implications and next steps for research, clinical practice, and/or public policy; one-paragraph conclusion. 

Supplementary materials  

Please note that supplementary materials are not checked and will be posted as supplied by the authors. Therefore, please double check. Please cite your Supporting Information as outlined here: https://journals.plos.org/plosmedicine/s/supporting-information - Please note you may use almost any description as the item name of your supporting information as long as it contains an "S" and number. For example, “S1 Appendix” and “S2 Appendix,” “S1 Table” and “S2 Table. Please ensure each supplementary material has a call out (link) from your main manuscript.

Comments from the reviewers:

Reviewer #1: The authors have addressed my minor comments. I have nothing further to add.

Reviewer #2: The revisions and the rebuttal letter have answered all my queries in a good and well-organised way. The article was good also before this round of revisions, but is now even better. I am also glad to see that the team has discussed the study and the findings with colleagues from SSA and invited them to take part in the work.

Nothing more to add from my side. Important work.

Reviewer #3: Thanks for revising the paper based on the feedback from the reviewers. There have been significant improvements to the methods of the paper, but here are some remaining issues I found.

1. Line 155-160: "In order to compute the SPEI at a timescale of 24 months (referred to as SPEI-24 hereinafter) for a given month during the study period, we first generated a time series by summing the water balance over the preceding 23 months up to the current month; then we transformed this series to a normal distribution with a mean of zero and standard deviation of one to eliminate the effect of both the local season and climate patterns."

This is still very confusing. I spent time reading the SPEI documentation. The documentation suggests that an essential step of fitting a probability distribution to the 24-month historical water balance data is missing here. This is integral for deriving the CDF used in the SPEI formula: SPEI = Z(CDF(WB)/(1-CDF(WB))). It is crucial to specify which probability distribution was selected to model the observed 24-month SPEI time series. Commonly, distributions such as the log-logistic, or Gamma, are employed. After fitting the appropriate distribution, the values are then transformed into a Z-score, reflecting a normal distribution with a mean of 0 and a standard deviation of 1. Given the prominence of SPEI as the major exposure metric in this study, it is imperative that the method of calculating this index is delineated with clarity and precision in the paper.

2. "..There should be no such bias in the current analysis because our start of "follow-up" and start of exposure are both the start of pregnancy (time zero). Although each fetus had to survive through the entire pregnancy to be eligible to be included in the study sample,…"

What you're describing here aligns exactly with the concept of immortal time bias. In the context of your study, the fetus is 'immortal' from conception to birth, yet this period is still included as exposure time in the analysis. Actually, I believe the exposure period should start at birth, as fetal demise is not considered child mortality anyway. I would highly recommend you change the exposure start time for all the analyses in the paper.

[LINK]

---

## [Decision Letter · Decision Letter 2]

5 Jul 2024

Dear Dr Wang,

Many thanks for submitting your manuscript "The drier, the deadlier—the association between long-term drought and risk of infant mortality in Africa: A cross-sectional study" (PMEDICINE-D-24-00825R2) to PLOS Medicine. The paper has been reviewed by subject experts and a statistician; their comments are included below and can also be accessed here: [LINK]

Thank you for your detailed response to the editors' and reviewers' comments. I have discussed the paper with my colleagues, and it has also been seen again by the statistical original reviewer. The changes made to the paper were mostly satisfactory. However, the editorial team concurs with the statistical reviewer that issue around survival bias needs to be resolved and implemented as suggested by the reviewer. Therefore, we ask you to carefully address the comments in a further revision to preclude the need for further major revisions and satisfy the reviewers and editors. When submitting your revised paper, please again include a detailed point-by-point response to the comments.

We ask that you submit your revision by Jul 26 2024 11:59PM. However, if this deadline is not feasible, please contact me by email, and we can discuss a suitable alternative.

Don't hesitate to contact me directly with any questions (kjanin@plos.org).

Best regards,

Katrien

Katrien Janin, PhD

Associate Editor

PLOS Medicine

kjanin@plos.org

Comments from the reviewers:

Reviewer #3: I am very confused by the authors' response regarding the survival bias, particularly concerning the time-zero of the study. Previously, the authors stated, "...There should be no such bias in the current analysis because our start of 'follow-up' and start of exposure are both the start of pregnancy (time zero)...". However, in the current response, they said, "...However, in our study, our exposure starts from the start of pregnancy, and the time zero is birth (time zero is LATER than the start of exposure)," which contradicts their earlier statement.

If time zero is indeed at birth, then the follow-up time should start at birth. In that case, there would be no survival bias, and that's why I suggested changing the time zero to birth (instead of the start of pregnancy) for their survival analyses. However, if time zero is the start of pregnancy, the period between the start of pregnancy and birth is immortal time. It seems the authors may not fully understand immortal bias. Immortal bias occurs when follow-up time is counted during a period when the outcome of interest cannot happen. Misalignment between time zero and treatment assignment can cause immortal bias, but this is not the definition of survival bias. If the authors start counting time from the the start of pregnancy, then there is survival bias. The authors seem to not fully understand the concepts, and I am very confused by their contradictory statements...

For clarification, here is a definition of time zero and follow-up time (time-to-event): "Time zero, or the time origin, is the time at which participants are considered at-risk for the outcome of interest... Follow-up time is measured from time zero (the start of the study or from the point at which the participant is considered to be at risk) until the event occurs, the study ends, or the participant is lost, whichever comes first." (https://sphweb.bumc.bu.edu/otlt/mph-modules/bs/bs704_survival/BS704_Survival_print.html#:~:text=individual%20being%20followed.-,Follow%20Up%20Time,(i.e.%2C%20at%20enrollment )

---

---

## [Decision Letter · Decision Letter 3]

10 Sep 2024

Dear Dr. Wang,

Many thanks for submitting your revised manuscript (PMEDICINE-D-24-00825R3) to PLOS Medicine. I’m contacting you on behalf of my colleague Katrien, who is no longer with the journal. The revised manuscript has been reviewed again by the statistical reviewer, and given the differing views of the original statistical reviewer and the authors, we also solicited the opinion of an independent statistician to contribute additional insight into the analysis and the question of immortal time bias. As you will see below (and attached), the original statistical reviewer and the arbitrating statistical reviewer (Reviewer 4) agree that the current statistical approach in the paper is problematic. Indeed, the arbitrating statistician does not feel that the analysis effectively accounts for the impact of the exposure (drought) during pregnancy (the reviewers’ comments can also be accessed here: [LINK]). More broadly, this reviewer does not feel that the conclusions as currently presented are sufficiently supported by the data.

In order to continue forward with the manuscript in view of the ongoing statistical questions, we must ask you to revised the paper again in response to the statisticians’ comments and the editorial points outlined below. Please note that the response and revisions must be satisfactory from a statistical standpoint for us to ultimately accept the paper for publication and that we will only consider one final revision.

As before, when you upload your revision, please include a point-by-point response that addresses all of the reviewer and editorial points, indicating the changes made in the manuscript and either an excerpt of the revised text or the location (eg: page and line number) where each change can be found. When you resubmit your paper, please include a clean version of the paper as the main article file and a version with changes tracked as a marked-up manuscript. It may also be helpful to check the guidelines for revised papers at http://journals.plos.org/plosmedicine/s/revising-your-manuscript for any that apply to your paper.

We ask that you submit your revision by October 1st. However, if this deadline is not feasible or you have any questions, please contact me directly by email (hvanepps@plos.org).

Kind regards,

Heather

Heather Van Epps, PhD

Executive Editor, PLOS Medicine

hvanepps@plos.org

[on behalf of]

Katrien Janin, PhD

Associate Editor

PLOS Medicine

kjanin@plos.org

Comments from Editors:

1. We appreciate the inclusion of the potential for immortal time bias as a limitation of the study. However, we feel that some additional consideration is needed in terms of the statistical approach given the stated importance of the pregnancy period as part of the exposure (noting that the editors agree with the notion that exposure to draught conditions during pregnancy could affect the developing fetus) and in view of the statistician’s comment that this exposure has not been accounted for in the model.

2. More detail is needed around the sensitivity analysis shown in figure S3, as noted by the arbitrating statistician.

3. Regarding the conclusions of the manuscript, I have discussed this with the editorial team, and we agree with the arbitrating statistical reviewer that the data presented in the paper do not support the strong conclusions. As noted by the statistician, in view of the results of the standard null hypothesis significance testing with a critical alpha of 0.05, the statement in the abstract should be amended to “Compared to children who did not experience drought, any drought exposure was *not* associated with an increased risk of infant mortality…” (or to the language suggested in the review). We also feel that the discordant result of the severe draught effect should be noted in the Abstract in the interest of full and transparent reporting.

4. The title of the manuscript should be changed to comply with PLOS Medicine style (it should not be declamatory) and to accurately reflect the data presented in the paper. We suggest “Long-term drought and risk of infant mortality in Africa: A cross-sectional study”.

5. We intend to send the revised paper to the statisticians for a final look. As noted above, it will be important for the paper to be acceptable from a statistical perspective for us to proceed to acceptance/publication.

Comments from the reviewers:

Reviewer #3:

I really appreciate the detailed explanations and the efforts the authors have made to revise the paper. However, I still do not agree with the decision to include the time between the start of pregnancy and the time of birth in the time-to-event outcome.

First, I fully agree with the authors' argument that "maternal exposure to drought during pregnancy is potentially associated with infant mortality through the mother." However, the time of exposure does not need to coincide with time-zero in a survival analysis. It can, but it does not have to. For example, you can examine the relationship between exposure to second-hand smoke before age 10 and "the risk of developing asthma after age 10." That's completely valid. However, the time-zero must start at age 10, not age 9, because the outcome is defined as "the risk of developing asthma <after> age 10." Any time before age 10 is "immortal time" because, by definition, it is impossible for the person to have the outcome.

For infant mortality, by definition, it is the death of a newborn <after birth> and before age 1. Therefore, "infant mortality" cannot occur before birth. Please note that the concept of time-zero in a survival analysis is related to the outcome (i.e., time to the event, which is infant mortality) rather than to the exposure. I believe the authors are still confused on this point.

It is encouraging to hear that using the time of birth as time-zero does not significantly change the results (in fact, by removing the immortal time period, I would expect the relationship to be stronger). Therefore, I recommend reporting those results using the time of birth as time-zero. Why do we have to introduce immortal time bias when it can be avoided?

Reviewer #4: See attachment

Michael Dewey

---

## [Decision Letter · Decision Letter 4]

6 Nov 2024

Hi Pin,

Thank you for getting back to me and apologies for the delay as I was out on vacation. Thank you also for updating the manuscript following our conversation. I have issued a major revision outcome so that you may upload your revised manuscript as well as any other supporting materials; the team will make a final decision based on this information and any additional consultation that we need.

Best wishes

Rebecca

Rebecca Kirk, MBA, PhD (she/her)

Associate Editorial Director

PLOS

---

## [Editor Report · Decision Letter 5]

12 Dec 2024

Dear Dr. Wang,

Thank you for re-submitting your manuscript "Long-term drought and risk of infant mortality in Africa: A cross-sectional study" (PMEDICINE-D-24-00825R5) and for your patience with what has been an unusually protracted process. We appreciate your continued willingness to engage with us and discuss the details of the statistical approach with the editors and statistical reviewers; this process has has resulted in increased clarity and improvements to the paper.

I have discussed the paper with my colleagues and one of the statistical reviewers, and I’m pleased to say that provided the remaining editorial and production issues are dealt with we are planning to accept the paper for publication in the journal. The remaining issues that need to be addressed are listed at the end of this email. Please take these into account before resubmitting your manuscript:

In revising the manuscript for further consideration here, please ensure you address the specific points made by the editors. In your rebuttal letter you should indicate your response to the reviewers' and editors' comments and the changes you have made in the manuscript. Please submit a clean version of the paper as the main article file. A version with changes marked must also be uploaded as a marked up manuscript file.

In view of the upcoming holidays, we ask that you submit the final revision by Monday, January 6th. If you have any questions in the meantime, please contact me directly at hvanepps@plos.org. Otherwise, we look forward to receiving the revised manuscript..

Kind regards,

Heather

Heather Van Epps, PhD

Executive Editor

PLOS Medicine

hvanepps@plos.org

Requests from Editors:

1. Abstract, line 67-68: please amend the wording from “…and a non-significant association with mild drought” to “but no significant association with mild drought”

2. Author summary: for full transparency, please include a bullet point to report the overarching finding that there was no evidence that any drought exposure was associated with an increased risk of infant mortality. It would make sense for this to precede the point about the association with severe drought.

3. Methods, line 163: please insert “et al” after the author name (Bendavid).

4. Results, line 276-278: please rephrase this sentence so that it’s consistent with the abstract. Ie, “Compared to children who did not experience drought, we did not find evidence that any drought exposure was associated with an increased risk of infant mortality (hazard ratio [HR]: 1.02; 95% confidence interval [CI] [1.00, 1.04], p=0.072).”

5. Results, line 281-283: please modify the following sentence to report lack of an association rather than a non-significant association. “A stronger but non-significant association with mild drought was estimated for postneonatal mortality (HR: 1.02; 95% CI [0.99, 1.06], p=0.203) than for neonatal mortality (HR: 0.99; 95 % CI [0.96, 1.02], p=0.657).”

6. Discussion: For completeness, we ask that you report the topline finding of no association between any drought exposure and infant mortality, as part of the initial paragraph of the Discussion.

7. References: please format the references according to PLOS style. In the text, references should be cited in square brackets (e.g., “We used the techniques developed by our colleagues [19] to analyze the data.” Full details on reference formatting can be found here: https://journals.plos.org/plosmedicine/s/submission-guidelines#loc-references.

8. Please remove the data availability statement from the main paper; this information will be included with the article meta-data (completed as part of the submission process).

9. Figure 1: Please confirm that the appropriate usage rights apply to the use of this map. Please see our guidelines for map images: https://journals.plos.org/plosmedicine/s/figures#loc-maps

10. Figure 2: please define all elements of box plots in the figure caption - center line, box limits and whiskers.

11. The supplemental file appears not to have been included in the most recent revision (these files are unfortunately not carried over from one revision to the next). Please ensure that a final version of the supplement is uploaded with this final revision.

---

## [Editor Report · Decision Letter 6]

19 Dec 2024

Dear Dr Wang, 

On behalf of my colleagues and the Academic Editor, Yuming Guo, I am pleased to inform you that we have agreed to publish your manuscript "Long-term drought and risk of infant mortality in Africa: A cross-sectional study" (PMEDICINE-D-24-00825R6) in PLOS Medicine.

PRESS

Thank you again for submitting to PLOS Medicine. We look forward to publishing your paper and hope that you will consider us for future publications.

Kind regards,

Heather

Heather Van Epps, PhD 

Senior Editor 

PLOS Medicine